# Temporal Stability of Ciliary Beating Post Nasal Brushing, Modulated by Storage Temperature

**DOI:** 10.3390/diagnostics13182974

**Published:** 2023-09-18

**Authors:** Noemie Bricmont, Romane Bonhiver, Lionel Benchimol, Bruno Louis, Jean-François Papon, Justine Monseur, Anne-Françoise Donneau, Catherine Moermans, Florence Schleich, Doriane Calmès, Anne-Lise Poirrier, Renaud Louis, Marie-Christine Seghaye, Céline Kempeneers

**Affiliations:** 1Pneumology Laboratory, I3 Group, GIGA Research Center, University of Liège, 4000 Liège, Belgium; rbonhiver@uliege.be (R.B.); c.moermans@chuliege.be (C.M.); fschleich@chuliege.be (F.S.); r.louis@chuliege.be (R.L.); ckempeneers@chuliege.be (C.K.); 2Division of Respirology, Department of Pediatrics, University Hospital Liège, 4000 Liège, Belgium; 3Department of ENT, University Hospital Liège, 4000 Liège, Belgium; lionel.benchimol@chuliege.be (L.B.); alpoirrier@chuliege.be (A.-L.P.); 4Institut Mondor de Recherche Biomédicale, INSERM-UPEC UMR 955, CNRS ERL7000, 94010 Créteil, France; bruno.louis@inserm.fr (B.L.); jean-francois.papon@aphp.fr (J.-F.P.); 5ENT Department, Assistance Publique-Hôpitaux de Paris (AP-HP), Université Paris-Saclay, Hôpital Bicêtre, 94270 Le Kremlin-Bicêtre, France; 6Biostatistics and Research Method Center-Public Health Department, University of Liège, 4000 Liège, Belgium; jmonseur@uliege.be (J.M.); afdonneau@uliege.be (A.-F.D.); 7Department of Pneumology, University Hospital Liège, 4000 Liège, Belgium; dcalmes@chuliege.be; 8Division of Cardiology, Department of Pediatrics, University Hospital Liège, University of Liège, 4000 Liège, Belgium; mcseghaye@chuliege.be

**Keywords:** cilia, ciliary beating, ciliary videomicroscopy, primary ciliary dyskinesia

## Abstract

Primary ciliary dyskinesia is a heterogeneous, inherited motile ciliopathy in which respiratory cilia beat abnormally, and some ultrastructural ciliary defects and specific genetic mutations have been associated with particular ciliary beating alterations. Ciliary beating can be evaluated using digital high-speed videomicroscopy (DHSV). However, normal reference values, essential to assess ciliary beating in patients referred for a PCD diagnostic, vary between centres, as minor variations in protocols might influence ciliary beating. Consequently, establishment of normal values is essential for each PCD diagnostic centre. We aimed to evaluate whether delay after sampling, and temperature for conservation of respiratory ciliated samples, might modify assessments of ciliary beating. In total, 37 healthy nasal brushing samples of respiratory ciliated epithelia were collected. Video sequences were recorded at 37 °C immediately using DHSV. Then, the samples were divided and conserved at 4 °C or at room temperature (RT). Ciliated beating edges were then recorded at 37 °C, at 3 h and at 9 h post sampling. In six samples, recordings were continued up to 72 h after sampling. Ciliary beating was assessed manually by ciliary beat frequency (CBF_M_) and ciliary beat pattern (CBP). A semi-automatic software was used for quantitative analysis. Both CBF and CBP evaluated manually and by a semi-automated method were stable 9 h after sampling. CBF_M_ was higher when evaluated using samples stored at RT than at 4 °C. CBP and the semi-automated evaluation of ciliary beating were not affected by storage temperature. When establishing normal references values, ciliary beating can be evaluated at 37 °C up to 9 h after nasal brushing, but the storage temperature modifies ciliary beating and needs to be controlled.

## 1. Introduction

The respiratory tract is lined by a mucociliary epithelium [1,2]. The main function of the respiratory epithelium is the clearance of mucus from the lower respiratory tract, which is the first line of defense of the lungs [1,3]. Mucociliary clearance results from an effective interaction between the mucus layer and normally coordinated ciliary beating. The highly coordinated beating of cilia allows the clearance of the mucus and entrapped inhaled pathogens and particles from the lower respiratory tract [1].

Primary ciliary dyskinesia (PCD) is a rare, inherited heterogeneous motile ciliopathy in which respiratory cilia are stationary, or beat in a slow or dyskinetic manner, leading to impaired mucociliary clearance and significant otosinopulmonary disease [4,5]. Ciliary beating can be evaluated using digital high-speed videomicroscopy (DHSV) [6,7,8]. DHSV allows evaluation of ciliary beat frequency (CBF) and beat pattern (CBP) [9]. PCD is an highly heterogeneous condition and some ultrastructural defects and genetic mutations have been associated with particular CBF and/or CBP alterations [6]. Mutations in genes encoding outer dynein arms components are associated with cilia that are mostly immotile or with residual movement (e.g., *DNAH5*, *CCDC151*) [2,10,11], with an exception of *DNAH11* mutations that present hyperkinetic and vibratile cilia [2]. Axonemal disorganization and absence of inner dynein arms (e.g., *CCDC39*, *CCDC40* mutations) have been linked with stiff cilia with reduced ciliary beat amplitude [2,11]. Combined inner and outer dynein arms defects were associated with an immotile CBP [5,10,11]. Abnormalities in the central microtubule complex and radial spoke head defects have been associated with a circular ciliary beating with normal CBF, uncoordinated ciliary beating, decreased ciliary beat amplitude, and immotile cilia [5,10,11].

DHSV is highly sensitive and specific for PCD diagnosis when performed by experienced investigators [12,13,14,15]. CBF should not be used without CBP assessment, given its lack of sensitivity and specificity to diagnose PCD [6,7,14]. Furthermore, minor variations among DHSV protocols might influence assessments of ciliary beating [16,17]. Given the lack of standard operating procedures for DHSV [6,7,10,12,13,18] and the heterogeneity of current DHSV protocols, normal reference values vary across centres. Normal values for ciliary beating evaluation are essential to be set up in each PCD diagnostic centre, in order to compare ciliary beating from samples obtained from patients referred for a PCD diagnostic.

Particularly, respiratory ciliated samples are stored before being processed for DHSV, and different storage durations and temperatures are reported in different laboratories [16]. Ciliated samples are sometimes processed for DHSV either directly after brushing or several hours after sampling [16], particularly if a sample is sent from a remote centre far from a PCD diagnostic centre.

However, it is unclear how ciliary beating might be modified by the delay after sampling or by the storage temperature for the conservation of ciliated samples prior to processing for DHSV, as previous data on healthy subjects have found conflicting results.

Sommer et al. [19] were the first to evaluate CBF time-dependent changes after sampling using DHSV at a controlled temperature of 21.5 °C. CBF was shown to increase gradually during the first 3 h after brushing, reaching a plateau between 3 h and 9 h after brushing, and then decreasing linearly.

Reula et al. [20] studied CBF and CBP changes linked to the storage duration and to the storage temperature (RT vs. 4 °C), using DHSV at room temperature. No significant differences were observed between samples conserved at RT or at 4 °C over time, in both CBF and the probability of dyskinetic CBP. Furthermore, the probability of dyskinetic CBP increased significantly with the storage duration up to 72 h after sampling, especially after 3 h. The same trend was observed for CBF; however, this was non-significant.

No studies evaluated the influence of the storage parameters on ciliary beating when samples are observed at 37 °C. However, as some temperature-sensitive variants of PCD may be missed if DHSV is performed under 37 °C [21], many DHSV laboratories assess ciliary beating at 37 °C.

In summary, data are discordant regarding the impacts of storage duration and temperature, and are lacking when DHSV is performed at 37 °C. The aim of this study was, thus, to evaluate if the storage temperature and duration should be taken into account within each DHSV laboratory when establishing normal values for ciliary function evaluated at 37 °C.

## 2. Materials and Methods

### 2.1. Study Design

Samples of respiratory ciliated epithelia were collected from the middle turbinate of 37 healthy subjects using a cytology brush and without local anaesthesia. Aiming to obtain good quality samples, nasal brushings were performed by 2 physicians trained and experienced in the technique, working within the PCD diagnostic centre of CHU of Liège. Exclusion criteria included chronic respiratory disease, familial history of PCD, respiratory tract infection during the previous 4 weeks, regular use of chronic nasal or inhaled medication, or active smoking. This observational study was approved by the ethics committee of CHU of Liège (2020-174), and written consent was obtained from all subjects.

Nasal brushing samples were placed in 2 mL of medium 199 (Thermo Fisher, Waltham, MA, USA) containing antibiotic solution (1% of penicillin/streptomycin (Thermo Fisher, Waltham, MA, USA)) and an antifungal (1% of amphotericin B (Thermo Fisher, Waltham, MA, USA)). Video sequences of ciliated beating edges were recorded using an inverted microscope with a x100 oil-immersion interference contrast objective (Axio Vert.A1, Zeiss, Oberkochen, Germany) and a high-speed video camera (CrashCam Mini 1510, IDT Innovation in motion, Pasadena, CA, USA), at a frame rate of 500 hertz (Hz) and at a controlled temperature of 37 °C. To record video sequences of beating cilia, 60 µL of respiratory ciliated edges within medium 199 were placed under the microscope and heated at 37 °C using an heated box (Ibidi, Gräfelfing, Germany) and a microscope lens heater (Tokai Hit, Fujinomiya, Japan), and the temperature was strictly controlled before each videorecording using a temperature probe, allowing to adjust the parameters of the heated box and the lens heater, as previously described [22].

Video sequences of beating cilia were recorded at 37 °C immediately after nasal brushing, then, for each subject, the ciliated samples within medium 199 were divided equally, and conserved either at 4 °C or at RT. Video sequences of ciliated beating edges conserved at 4 °C and at RT were recorded under microscope at 37 °C, at 3 h (H3), then at 9 h (H9) after sampling. In a subgroup of 6/37 healthy subjects, nasal brushing samples were conserved up to 72 h at 4 °C and at RT, and ciliated beating edges were recorded at 37 °C, at 24 h (H24), 48 h (H48) and 72 h (H72) after sampling.

Only normal edges or edges with minor projections [23], at least 50 µm in length, were recorded and used for ciliary functional analysis (CFA). Within these edges, only cilia free of mucus and those beating in the sideways profile were analysed. CFA was evaluated from a minimum of 3 high quality edges meeting the above criteria, for each time and storage temperature condition. Nasal brushing samples that did not allow a CFA at H0, H3 and H9 were excluded.

### 2.2. Manual Ciliary Functional Evaluation

To perform the manual CBF (CBF_M_) evaluation, cilia or groups of cilia beating in the sideways profile were identified, the number of frames required to complete 5 beat cycles was counted, and this was converted to CBF by a simple calculation [9]. A maximum of 10 CBF_M_ measurements were calculated from each ciliated beating edge. Ciliated edges that did not allow a minimum of 4 CBF_M_ measurements along the edge were excluded from CFA. If immotile cilia were observed, a CBF of 0 Hz was recorded. For each sample, the mean CBF was calculated at each time-point and storage temperature condition.

The precise path taken by a cilium or group of cilia during a complete beat cycle was compared to normal CBP, observed using DHSV [9,24]. For each cilium or group of cilia used for a manual CBF evaluation, a distinct normal or abnormal CBP was assigned, and the percentage of abnormal CBP within the sample was calculated for each time-point and storage temperature condition.

### 2.3. Computer-Assisted Ciliary Functional Evaluation

The semi-automatic software used for quantitative analysis of ciliary beating was developed by the research team of the PCD Diagnostic Centre in Paris [15]. This software allows the calculation of 12 objective parameters by determining three localization points characterizing the complete cycle of each cilium, and five time-points, as previously described [15].

Among these 12 parameters, three were selected: CBF obtained by this semi-automatic method (CBF_A_), ciliary beat amplitude (CBA) and distance travelled by the cilium per second weighted by the percentage of beating ciliated edges (DTSW) [15,25]. CBF_A_ was selected because it should be equivalent to CBF_M_. CBA was selected because it allows characterization of CBP. DTSW was selected because, among these parameters, it was found to be the most discriminating for distinguishing between PCD and no PCD [15].

### 2.4. Statistical Analysis

The normality of the distribution of the quantitative variables was investigated with mean and median values, histogram, quantile–quantile plot, and Shapiro–Wilk normality test. All CFA parameters collected during the study had a symmetrical distribution and were described using the mean and standard deviation (±SD). The evolution of each CFA parameter with variation in storage duration and temperature was assessed using a mixed linear model adjusted for the considered parameter at the time of sampling. In addition, each parameter was compared between H0 and H3 using Student’s *t* test for paired data depending on temperature of storage.

No sample size calculation was carried out prior to conducting the study; analyses were performed on the maximum number of healthy subjects available (*n* = 26). However, the available sample size made it possible to reject equality of means for differences corresponding to a moderate effect size (0.57) with a power of 80%. Statistical software used were SAS version 9.4 for Windows (SAS Institute, Cary, NC, USA) and R version 4.2 (R Core Team, Vienna, Austria). Significance was assessed at level 5% (*p* < 0.05).

## 3. Results

### 3.1. Manual CFA over 9 Hours

Of the 37 nasal brushing samples collected, 26 samples (21 females and 5 males, median age 25.0 (24.0–31.0) years) met the defined quality criteria and allowed a manual CFA evaluation (including CBF_M_ and the percentage of abnormal CBP) at H0, then after storage at both 4 °C and at RT, at H3 and at H9 after sampling (Figure 1).

CBF_M_ was stable between H0 and H3 after sampling at 4 °C and at RT (*p* = 0.56 and *p* = 0.18, respectively) (Table 1). The mixed linear model showed that there was no interaction between time and storage temperature (*p* = 0.44), as CBF_M_ followed the same evolution over time for both storage temperature conditions (Figure 2). Moreover, CBF_M_ was not significantly affected by storage duration (*p* = 0.54). However, CBF_M_ was significantly higher in samples stored at RT compared to samples stored at 4 °C (coefficient: 1.11 ± 0.51 Hz, *p* = 0.041). This difference in CBF_M_ between storage conditions remained unchanged regardless of the duration of conservation (*p* = 0.44) (Figure 2).

Similarly, the percentage of abnormal CBP was stable between H0 and H3 after sampling at 4 °C and at RT (*p* = 0.55 and *p* = 0.69, respectively) (Table 1). The mixed linear model showed no interaction between the duration of storage and the storage temperature (*p* = 0.58), and showed the percentage of abnormal CBP was not significantly affected by the storage duration (*p* = 0.70) (Figure 3). However, contrary to the evolution of CBF_M_, the percentage of abnormal CBP was not significantly affected by storage temperature (*p* = 0.59) (Figure 3).

### 3.2. Computer-Assisted CFA over 9 Hours

From the 26 nasal brushing samples used to evaluate manually CFA at H0, H3 and H9, a subgroup of 10 samples (median age, 25.5 (23.8–32.8) years) was used to evaluate CBF_A_, CBA and DTSW using the semi-automatic software at H0, H3 and H9 (Figure 1).

These three parameters were stable between H0 and H3 after sampling at both 4 °C and RT (Table 1). The mixed linear model also showed no interaction between storage duration and storage temperature, as CBF_A_, CBA and DTSW presented the same evolution over time for each storage temperature condition (*p* = 0.94; *p* = 0.84 and *p* = 0.97, respectively) (Figure 4A–C). In addition, these three parameters were not significantly affected by the storage duration (*p* = 0.58; *p* = 0.37; *p* = 0.37; respectively) or by the storage temperature (*p* = 0.39; *p* = 0.82 and *p* = 0.99, respectively) (Figure 4A–C).

### 3.3. Manual CFA over 72 Hours

From the 26 nasal brushing samples used to evaluate CFA at H0, H3 and H9, a subgroup of six samples (median age, 24.5 (22.8–31.8) years) was evaluated for CBF_M_ and the percentage of abnormal CBP up to 72 h after sampling (Figure 1).

The mixed linear model showed that CBF_M_ remained significantly higher in samples stored at RT compared to samples stored at 4 °C, when evaluated up to H72 (coefficient: 1.47 ± 10.52 Hz, *p* = 0.037) (Figure 5). The mixed linear model also showed that CBF_M_ was stable over time (*p* = 0.33). Nevertheless, a significant decrease between H3 and H72 was observed (*p* = 0.042) (Figure 5).

The mixed linear model also showed that the storage duration had no statistically significant influence on the percentage of abnormal CBP, when evaluated up to 72 h after sampling (*p* = 0.44) (Figure 6). However, contrary to the results obtained when evaluating samples up to H9, the mixed linear model showed that the percentage of abnormal CBP evaluated up to 72 h after sampling was significantly higher in samples stored at RT compared to samples stored at 4 °C (coefficient: 7.99 ± 2.84%, *p* = 0.037) (Figure 6).

## 4. Discussion

Our results showed that both CBF and CBP, evaluated at 37 °C from healthy ciliated samples by a manual and by a semi-automated method, are stable for 9 h after sampling, at storage temperatures of 4 °C and RT. The extended evaluation up to 72 h after sampling in a subset of six ciliated samples showed a decreasing trend in CBF_M_ between 9 h and 24 h, for samples conserved at 4 °C only (Figure 5).

We also showed that CBF_M_ is significantly higher when evaluated using samples stored at RT compared to those stored at 4 °C. The percentage of abnormal CBP and the ciliary beating evaluated by a semi-automated method were not affected by the storage temperature when evaluated up to 9 h after sampling. However, when evaluation time was extended up to 72 h, the percentage of abnormal CBP was higher in samples conserved at RT compared to 4 °C.

Therefore, our results suggested that CFA evaluated by DHSV at 37 °C might be performed within 9 h after sampling. This might be extended up to 48 h for samples conserved at RT, taking into account that these data have been obtained from only six samples. The results also suggested that the storage temperature might influence ciliary beating, and should be considered when establishing normative data.

However, these results differed from previous studies evaluating the influence of delay and storage temperature on ciliary beating. Two studies evaluated ciliary beating at RT in healthy ciliated samples and found differing results, and recommended the evaluation of ciliary beating either within 3 h after sampling [20], or between 3 h and 9 h after sampling [19]. Only one paper assessed the influence of storage temperature on ciliary beating, and found no difference between samples stored at RT or at 4 °C [20].

While our results demonstrated a stability of CBF up to 9 h after sampling, Sommer et al. [19] described a CBF = 5.7 ± 2.5 Hz at H0, increasing up to 3 h after sampling, with stability reached between H3 and H9, followed by a CBF decrease, and Reula et al. [20] reported CBFs = 11.99 ± 3.34 Hz (RT) and 10.79 ± 1.80 Hz (4 °C) at H0, with an increase over time, up to 72 h after sampling. These discrepancies in results might be explained by the differences in the temperatures used in DHSV protocols to evaluate ciliary beating (at a controlled temperature of 21.5 °C in the Sommer et al. [19] study, at an uncontrolled RT in the Reula et al. [20] study, and at a controlled temperature of 37 °C in our study). Indeed, previous data have shown that CBF varies with temperature, and that this relationship is sigmoidal [26]. This likely explains the variability in CBF at H0 in the three studies, and might explain a higher variability in CBF when evaluated at 21.5 °C or at RT than at 37 °C, as minor variations in temperature induce wider CBF changes at this temperature range. Furthermore, the temperature used to visualize cilia is reported as non-controlled in the Reula et al. [20] study, and controlled in the Sommer et al. [19] study. However, in our experience, it is difficult to maintain an exact temperature for the entire duration of an experiment.

Furthermore, we found that CBP, evaluated both manually and using a semi-automated method, was stable up to 72 h after sampling. However, Reula et al. [20] described that the probability of CBP being dyskinetic increased significantly with storage duration up to 72 h, especially after H3. Interestingly, the percentage of abnormal CBP at H0 in samples from healthy subjects was much higher in the Reula et al. [20] study (47.10% and 57.10% for samples stored at RT or at 4 °C) than in our study (19.81 ± 9.52%). The lower percentage of dyskinesia found in our study might also be explained by the selection of ciliated beating edges to evaluate ciliary beating. Indeed, Thomas et al. [23] demonstrated that disrupted ciliated epithelial edges showed a lower CBF and an increased dyskinesia.

Finally, we observed that CBF measured at 37 °C was higher when evaluated from samples stored at RT, compared with samples stored at 4 °C. Previous studies showed that CBF measured at a given temperature is higher when the sample had been gradually heated than when it had been cooled [26,27]. Our study additionally suggested that CBF evaluated at 37 °C was higher when heated from a higher starting temperature, RT, than when heated from a lower starting temperature, 4 °C.

To reduce the subjectivity of the manual evaluation of CBP, in our study, the evaluation of the stability of the ciliary function was repeated using semi-automatic software. This confirmed the results obtained via the manual method: ciliary function was stable over time, regardless of the storage temperature of the samples. However, unlike CBF_M_, CBF_A_ was not significantly affected by storage temperature. This was likely due to a lower number of samples used. This might also be explained by the difference in the methods used for CBF evaluation, as the semi-automated software did not include immotile cilia in the CBF calculation as did the manual method. However, no significant differences were observed when comparing the manually assessed percentage of immotile cilia within the subgroup of 10 subjects included in the semi-automated analysis, for samples stored at 4 °C or at RT, 3 h after sampling (0% (0–4.63) vs. (0% (0–8.08), respectively, *p* = 0.31) and 9 h after sampling (0% (0–5.00) vs. 0% (0–4.90), respectively, *p* > 0.99).

## 5. Conclusions

Our results suggest that, in healthy subjects, both CBF and CBP can be evaluated within 9 h after nasal brushing, regardless of the storage temperature (in the fridge or at room temperature), when ciliary beating is evaluated at a controlled temperature of 37 °C. Ciliary beating evaluation might be extended up to 48 h for samples conserved at RT only, but this should be evaluated in a larger cohort. CBP evaluation is required in PCD diagnosis [6]; however, manual CBP evaluation is subjective. We confirmed the stability of CBP over time up to 9 h after brushing, using a semi-automated software to quantify CBP evaluation. Furthermore, our results suggest that the storage temperature modifies ciliary beating, and this must be considered when establishing normal values in a laboratory or to when comparing results obtained in different laboratories. However, this study evaluated the stability of ciliary function in healthy subjects only, and these results should be confirmed in a larger cohort including patients referred to a PCD diagnostic centre, with a positive or negative diagnostic result. Indeed, chronic bacterial infection and the chronic inflammatory environment within the respiratory tract in these patients may affect ciliary beating and its stability over time. Furthermore, PCD is a highly heterogeneous condition and some ultrastructural defects and genetic mutations have been associated with particular CBF and/or CBP alterations. Consequently, a study aiming to assess the stability of ciliary function in PCD should include PCD patients with genetic mutations associated with different ciliary beating abnormalities, and different ultrastructural defects.

## Figures and Tables

**Figure 1 diagnostics-13-02974-f001:**
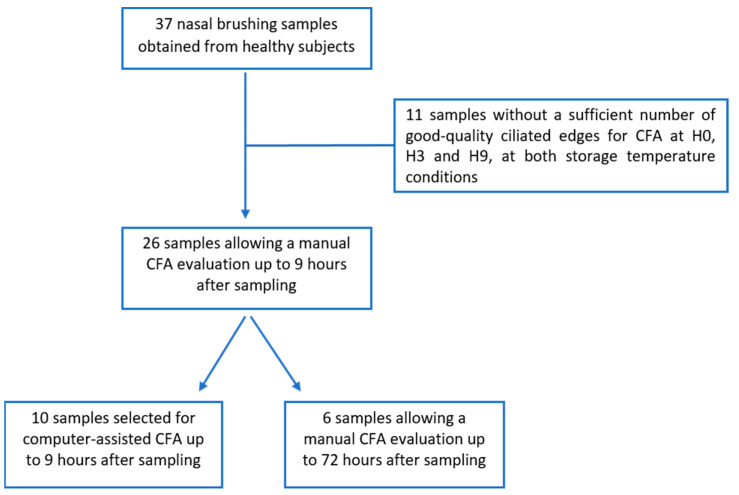
Flowchart: 37 nasal brushing samples were obtained from healthy subjects. 11/37 samples did not allow to obtain enough ciliated edges corresponding to our quality criteria to perform ciliary functional analysis (CFA) at H0, H3, and H9, at both storage temperature conditions (room temperature and 4 °C). From the 26 samples that allowed a manual CFA at H0, H3, and H9 after sampling, the 10 first samples included were also used to evaluate ciliary beating using a computer-assisted method at H0, H3, and H9. Only 6/26 samples contained enough ciliated edges to perform a manual CFA at H0, H3, H9, H24, H48, and H72, at both storage temperature conditions. CFA = ciliary functional analysis.

**Figure 2 diagnostics-13-02974-f002:**
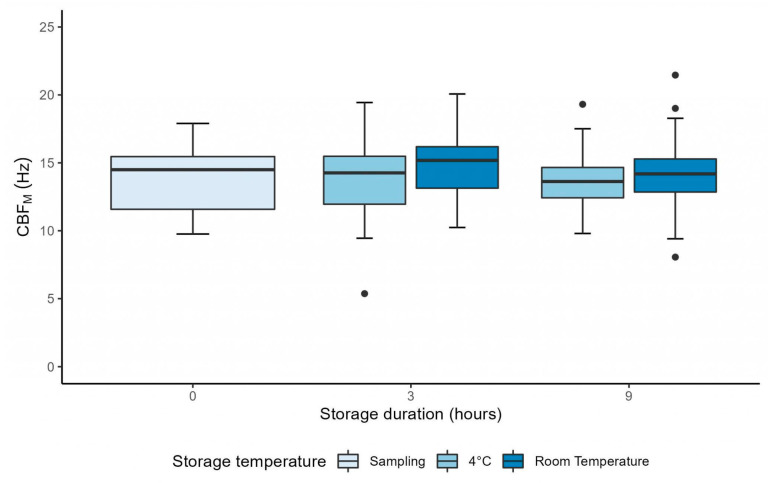
Graph showing the evolution of manual ciliary beat frequency up to 9 h after sampling, in samples stored at 4 °C and at room temperature. The difference between 4 °C and room temperature was significant (mixed linear model, *p* = 0.041). CBF_M_ = manual ciliary beat frequency; Hz = Hertz.

**Figure 3 diagnostics-13-02974-f003:**
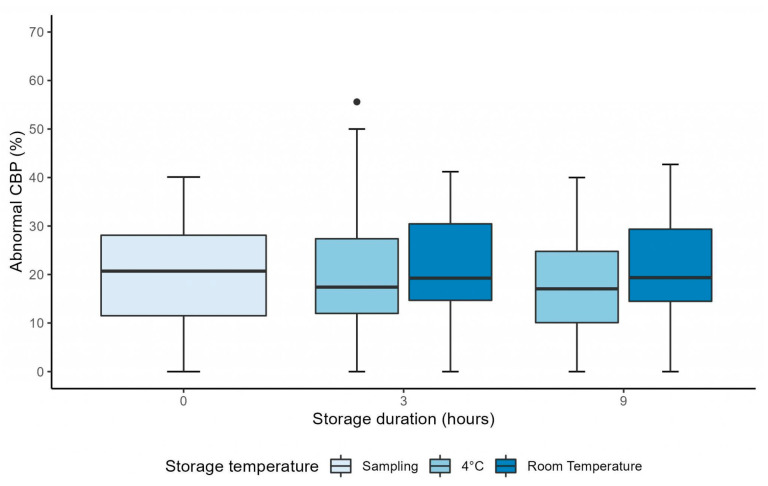
Graph showing the evolution of the percentage of abnormal ciliary beat pattern up to 9 h after sampling, in samples stored at 4 °C and at room temperature. CBP = ciliary beat pattern.

**Figure 4 diagnostics-13-02974-f004:**
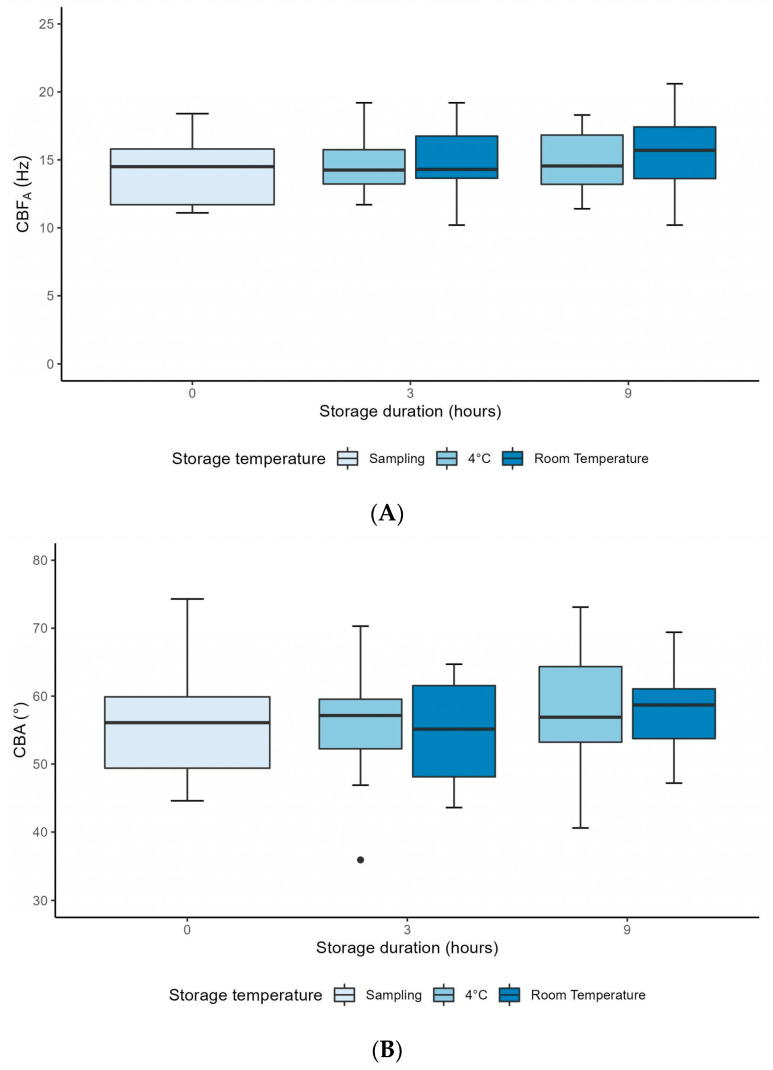
Graphs showing the evolution of the semi-automatic ciliary beat frequency (**A**), the ciliary beat angle (**B**) and the distance travelled by the cilium per second (**C**), up to 9 h after sampling, in samples stored at 4 °C and at room temperature. CBF_A_ = semi-automatic ciliary beat frequency; CBA = ciliary beat angle; Hz = Hertz.

**Figure 5 diagnostics-13-02974-f005:**
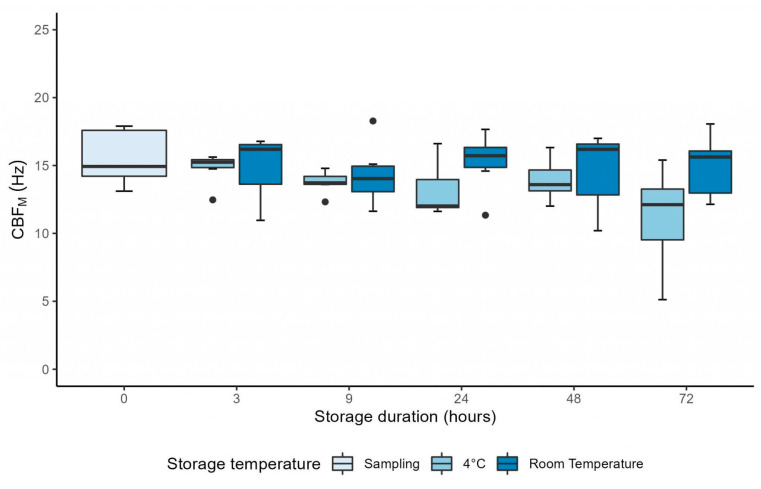
Graph showing the evolution of manual ciliary beat frequency up to 72 h after sampling, in samples stored at 4 °C and at room temperature. The difference between 4 °C and room temperature was significant (mixed linear model, *p* = 0.037) and the difference between H3 and H72 was significant (mixed linear model, *p* = 0.042). CBF_M_ = manual ciliary beat frequency; Hz = hertz.

**Figure 6 diagnostics-13-02974-f006:**
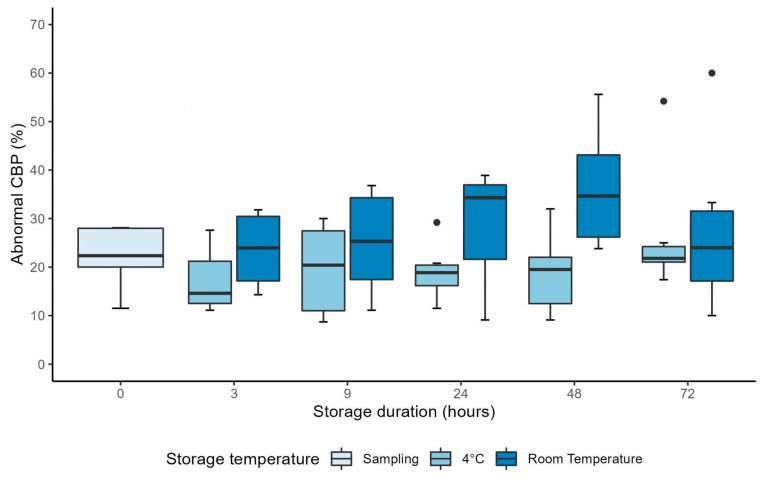
Graph showing the evolution of the percentage of abnormal ciliary beat pattern up to 72 h after sampling, in samples stored at 4 °C and at room temperature. The difference between 4 °C and room temperature was significant (mixed linear model, *p* = 0.037). CBP = ciliary beat pattern.

**Table 1 diagnostics-13-02974-t001:** Description and comparison of the manual and semi-automatic ciliary beat frequency, the percentage of abnormal ciliary beat pattern, the ciliary beat angle and the distance travelled by the cilium per second up to 3 h after sampling in samples stored at 4 °C and at room temperature.

		H0	H3	*p*-Value
CBF_M_ (Hz) (*n* = 26)	4 °C	14.01 ± 2.36	13.65 ± 2.90	0.56
	RT		14.76 ± 2.17	0.18
Percentage of abnormal CBP (%) (*n* = 26)	4 °C	19.81 ± 9.52	21.40 ± 14.09	0.55
	RT		20.95 ± 11.48	0.69
CBF_A_ (Hz) (*n* = 10)	4 °C	14.33 ± 2.45	14.56 ± 2.16	0.83
	RT		15.03 ± 2.68	0.42
CBA (°) (*n* = 10)	4 °C	56.08 ± 8.59	55.80 ± 9.59	0.94
	RT		54.58 ± 7.86	0.69
DTSW (µm) (*n* = 10)	4 °C	70.73 ± 14.23	76.63 ± 21.69	0.47
	RT		77.04 ± 18.33	0.30

CBA: ciliary beat angle; CBF_M_: manual ciliary beat frequency; CBF_A_: semi-automatic ciliary beat frequency; CBP: ciliary beat pattern; DTSW: the distance travelled by the cilium per second weighted by the percentage of beating ciliated edges; Hz: hertz; RT: room temperature. Results are expressed as mean ± SD. Means were compared with a Student’s paired *t*-test.

## Data Availability

All data is contained within the article.

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
