# Peer review of "Temporal Stability of Ciliary Beating Post Nasal Brushing, Modulated by Storage Temperature"

_diagnostics, 2023, doi:10.3390/diagnostics13182974_

Round 1

Reviewer 1 Report

The work presented for evaluation concerns the analysis of ciliary beating frequency and beat pattern of respiratory epithelial cilia. The analysis was carried out under different laboratory conditions (different temperatures and time points). The results obtained seem to be interesting and clinically useful.

The introduction of the paper deals with the abnormalities observed in PCD, while the whole analysis is based on epithelium obtained from healthy controls.  Material, methods, and results do not correspond to the content of the introduction and abstract. This is an inconsistency that may confuse the reader. 

 I suggest:

- amend the introduction, describe the structure and function of the respiratory epithelium, and the role of mucociliary clearance in the respiratory tract.  Briefly describe the abnormalities of CBF and CBP in PCD, depending on the mutation.

- it is necessary to change the content of the abstract.

Reviewer 2 Report

Dear Authors,

I have carefully reviewed your manuscript titled "Ciliary beating remains stable 9 hours after nasal brushing, but is modified by storage temperature" and appreciate the effort you have put into investigating the impact of time delay and temperature on ciliary beating patterns. However, I have identified a significant limitation in your study design that warrants major revisions before the manuscript can be considered for publication. The absence of a positive control group and the exclusive use of healthy controls undermine the generalizability and clinical relevance of your findings as you mention in the introduction, we conduct DHSV as part of the diagnostic evaluation in patients with PCD. The fact that PCD patients were not included overlooks the crucial aspect of comparing ciliary beating patterns between healthy controls and those diagnosed PCD. To establish the clinical significance of your findings, it is imperative to include PCD patients as a positive control group. This would allow you to compare ciliary beating patterns between healthy individuals and those with a known ciliary dysfunction, highlighting any substantial differences that could be attributed to the condition. The absence of PCD patients in your study restricts the scope of its implications and applicability. Without a positive control group, it becomes challenging to ascertain whether the observed alterations in ciliary beating patterns are solely due to the storage conditions or if they could potentially be influenced by inherent differences between healthy individuals and those with ciliary disorders.

Sample Size: Adequately power your study to account for the inclusion of PCD patients. This may require adjustments to your initial sample size calculation.

Data Analysis: Perform a comprehensive analysis to compare ciliary beating patterns between healthy controls and PCD patients. This will provide valuable insights into the clinical relevance of your findings.

Title: Consider a new title like: “Temporal Stability of Ciliary Beating Post Nasal Brushing, Modulated by Storage Temperature.” This title provided aims to encapsulate the essential findings and implications of your study in a more concise and informative manner.

In summary, incorporating a positive control group of PCD patients is crucial to enhancing the clinical significance and generalizability of your study's findings. By addressing this limitation, your manuscript will contribute more effectively to the understanding of ciliary dysfunction and its implications. I look forward to reviewing the revised version of your manuscript once these major revisions have been implemented.

Moderate editing of English language required

Reviewer 3 Report

This is a nicely written protocol of various laboratory practices used in DHSV for ciliary functional analysis. The authors used a rigorous methodological approach to this protocol evaluating CFA at various temperatures and after various periods of storage, which approximates the real-world barriers for CFA in clinics which may be at considerable distance from the expert lab. I have only minor critiques that the authors should clarify to improve the applicability of this manuscript.

1.  Were all nasal ciliary biopsies done by the same person or by various people? This is important as quality of samples can be greatly affected by inexperience of the operator obtaining cellular samples.

2. After sample storage at RT or 4 degrees, were samples re-warmed to 37 degrees for analysis at all time points? This is not clear. If they were rewarmed, was there a standardized re-warming protocol and time of re-warmth used in all samples?

3.       If samples were re-warmed for analysis and re-cooled for storage multiple times for the 3, 9, 72 hour time points, can the authors comment on how this repeated process of temperature fluctuations may have influenced their data (is there anything previously known on this process affecting CFA)?

4.       More explanation in figure 1 is needed, within the figure itself or in the caption. Additions that I think would be helpful include:

a.       More information (at least a box in the figure) describing numbers and reasons for sample exclusion.

b.       I see that the first 10 samples were sent for computer-assisted CFA at 72 hours were chosen at random, but how were did author choose which samples (n=10) to send for computer-assisted CFA at 9 hours? If not chosen at random, authors need to speculate on how this may have influenced results.

5.       The results at bottom of page 4, what does the abbreviation “CBFms” mean?

6.       The results at bottom of page 4, there is a significant p value of 0.041 that I do not see anywhere in the figure 2. Authors also claim significant differences in data in Figure 2, but all of the box and whisker plots look non-significant to me. Can authors highlight the statistically significant differences within figure 2?

7.       Similar to the comment above, can authors please highlight statistical difference in all figures, as only figure 5 shows obvious differences to me? Otherwise, the figures are very difficult for a reader to appreciate the significant differences.

8.       Many samples failed to even reach analysis in this world-expert center, which is a major limitation for this process in the clinical world, and this requires discussion. If a single expert collected nasal cells, this is even more significant as it demonstrates the limitations of DHSV even in world-expert hands.  

9.       While the authors do mention in the conclusion that this process must be done in PCD patients to see if results will be similar, I think more discussion on this point is needed. Versus healthy controls used in this protocol, cells of PCD patients will be affected by chronic bacterial infections and the inflammatory milieu will be completely different. These issues may greatly affect results in PCD patients, and this manuscript must discuss those potential large limitations when this is expanded to real-world clinical scenarios, and delayed processing of DHSV samples may not be appropriate in diseased patients.

only minor edits for English language and grammar are needed, 

Round 2

Reviewer 1 Report

I accept publication in present form.

Reviewer 2 Report

All my comments were addressed